# The Effect of Ice-Nucleation-Active Bacteria on Metabolic Regulation in *Evergestis extimalis* (Scopoli) (Lepidoptera: Pyralidae) Overwintering Larvae on the Qinghai-Tibet Plateau

**DOI:** 10.3390/insects13100909

**Published:** 2022-10-07

**Authors:** Hainan Shao, Yunxiang Liu, Yujiao Liu, Youpeng Lai

**Affiliations:** 1State Key Laboratory of Plateau Ecology and Agriculture, Qinghai University, Xining 810000, China; 2Key Laboratory of Agricultural Integrated Pest Management of Qinghai Province, Qinghai Academy of Agriculture and Forestry Sciences, Xinning 810000, China

**Keywords:** INA bacteria, *Evergestis extimalis*, supercooling point, trehalose

## Abstract

**Simple Summary:**

*Evergestis extimalis* (Scopoli) causes a serious reduction in the yield of spring oilseed rape every year. Until now, using chemosynthesis pesticides has been the main strategy to control *E. extimalis*. Biopesticide, as an alternative treatment, has attracted widely attention as it has little effect on the natural enemy and a low potential for promoting resistance. Our results have indicated that ice-nucleation-active microbes were able to stimulate trehalase activity, thereby leading to the reduction of trehalose accumulation. Consequently, *E. extimalis* showed an obviously decreased survival rate and ability to withstand low temperatures in the Qinghai-Tibet Plateau. Therefore, the use of INA bacteria has the potential to become a new strategy for the biological control of pests.

**Abstract:**

*Evergestis extimalis* (Scopoli) is a significant pest of spring oilseed rape in the Qinghai-Tibet Plateau. It has developed resistance to many commonly used insecticides. Therefore, biopesticides should be used to replace the chemical pesticides in pest control. In this study, the effects of ice-nucleation-active (INA) microbes (*Pseudomonas syringae* 1.7277, *P. syringae* 1.3200, and *Erwinia pyrifoliae* 1.3333) on *E. extimalis* were evaluated. The supercooling points (SCP) were markedly increased due to the INA bacteria application when they were compared to those of the untreated samples. Specifically, the SCP of *E. extimalis* after its exposure to a high concentration of INA bacteria in February were −10.72 °C, −13.73 °C, and −14.04 °C. Our findings have demonstrated that the trehalase (*Tre*) genes were up-regulated by the application of the INA bacteria, thereby resulting in an increased trehalase activity. Overall, the INA bacteria could act as effective heterogeneous ice nuclei which could lower the hardiness of *E. extimalis* to the cold and then freeze them to death in an extremely cold winter. Therefore, the control of insect pests with INA bacteria goes without doubt, in theory.

## 1. Introduction

Pesticides have played key roles in agriculture systems worldwide in the past century, allowing for a significant increase in food production and crop yields. Notwithstanding, the rapidly growing human population further stresses the need for an increased food production. Approximately 3.5 million tonnes of pesticides are used annually worldwide. Pesticide residues have given rise to various environmental problems, including poisoning human foods, and severely polluting terrestrial ecosystems as well as aquatic systems around the world [1,2]. Therefore, biopesticides should be used to improve crop yield because they are more specific to the targets and not prone to producing insecticide resistance when they are compared with chemical pesticides. Microbial pesticides, which are used to control insects, grasses, and diseases, are composed of bacteria, fungi, and viruses, and their metabolites [3]. Ice-nucleation-active (INA) microbes were discovered in *Erwinia*, *Pseudomonas,* and *Xanthomonas* [4,5,6], and they are in most INA bacteria which are normally found growing on plants, with only a few being known to come from frog and insect guts [7,8]. Some researchers have reported that INA bacteria could control insect pests [3,6,9]. INA bacteria could act as effective heterogeneous ice nuclei which can increase an insects’ supercooling point (SCP), lower their hardiness to the cold, and then freeze them to death after a low-temperature treatment. Therefore, there is no doubt that INA bacteria can control insect pests [10].

*Evergestis extimalis* (Scopoli) is primarily distributed in the eastern agricultural area and also occurs in the region that is along the coast of the Yellow River. *E. extimalis* is mainly fed on fennel, beet, cabbage, rape, radish, and kale. Its larvae usually damage oilseed rape crops by drilling through the pods and feeding on its their seeds. Zhang et al. [11,12] and Lai et al. [13] reported that the outbreak of *E. extimalis* in the Huangyuan and Pingan counties caused 60–100% of the yield loss of oilseed rape, thereby resulting in an enormous economic loss as oilseed rape is the most important oil alpine crops in the Qinghai Province where they are extensively grown. Lai et al. [14] reported that *E. extimalis* have developed a hardiness to the cold and its damage had reached crops that are located 2800 m above sea level. According to a study that was conducted in 2014, the death rate of the over-wintering larvae would not exceed 50% in the cold regions with an altitude of 3010 m, indicating that its supercooling capacity can adapt to the low-temperature stress of a local winter [15]. Furthermore, with the widespread and abused use of the pesticides, the resistance to *E. extimalis* has developed rapidly (unpublished data). Therefore, there is an urgent need for new, efficient biological control agents to overcome the problems that are caused by chemical insecticides. 

The Qinghai-Tibet Plateau is characterized by a high altitude and cold temperatures which pose a huge threat to the survival of the insects. Low winter temperatures limit the insect populations, activities, and development. The ability to survive annual temperature minima could be a key determinant of insect distribution [16]. A high level of trehalose in the body plays a key role in protecting them against chilling stresses [17,18]. For example, *Drosophila immigrans* and *D. melanogaster* that are hardened or acclimated to cold stress are characterized by a stress-triggered accumulation of trehalose [16,17,19]. Furthermore, alterations in the level of trehalose could regulate the growth and cold tolerance of *Trichogramma dendrolimi* [16]. Trehalase (EC 3.2.1.28) hydrolyzes one trehalose molecule into two glucose molecules in insect species, many of which have been identified and cloned, including in *Bombyx mori* (Lepidoptera: Bombycidae) [20] and *Glyphodes pyloalis* Walker (Lepidoptera: Pyralidae) [21]. However, the effect of an INA treatment on the expression pattern of trehalase in *E. extimalis* has not been reported. Given the wide distribution of *E. extimalis*, the studies have mainly focused on the biological control and pheromones of it, but the research on the biochemical and molecular mechanisms underlying the effect of INA bacteria on *E. extimalis* is lacking. This study is aimed to determine whether INA bacteria could decrease the cold tolerance of *E. extimalis*, and influence trehalose metabolism as well as the expression of the key metabolic enzymes.

## 2. Materials and Methods

### 2.1. Experimental Insects and Bacterial Strains

*Evergestis extimalis* (Scopoli) larvae and adults were collected from the mature oilseed rape in September in 2014, in the Xibao Town of Huangzhong County (Longitude: 101.35 °E; Latitude: 36.31 °N). The insects were reared on oilseed rape and kept in a growth chamber at 25 ± 1 °C and a 70 ± 10% relative humidity with a photoperiod of 14: 10 h (light:dark). *Pseudomonas syringae* 1.3200, *P. syringae* 1.7277, and *Erwinia pyrifoliae* 1.3333 were kindly provided by the Institute of Microbiology, Chinese Academy of Sciences and maintained at −80 °C before their use. The frozen culture was propagated in TSBYE at 37 °C for 24 h before its use. The resulting cultures were washed twice using sterile phosphate-buffered saline (PBS, pH 7.4), and the bacteria were resuspended in sterile PBS. The concentrations of the bacterial suspension were adjusted to approximate 5.0 × 10^7^, 1.0 × 10^8^, and 1.5 × 10^8^ CFU/mL and the optical densities of (OD) (C1(0.1), C2(0.2), C3(0.3)) at 600 nm were measured.

### 2.2. Cold Resistance of Overwintering Larvae

Firstly, a total of 100 mesh bags were used. After being carefully collected, the mature larvae (n = 20) of *E. extimalis* were put into a mesh bag (10 cm × 20 cm) with 0.5 kg of sieved soil. The mesh bag was buried in the experimental field (Plant Protection Research Institute, Qinghai Academy of Agriculture and Forestry Sciences). In late October in 2014, the mesh bags were dug up and the living cocoons were picked out. Secondly, a total of 1200 cocoons (Figure 1D) were used and sprayed with 150 mL of bacterial diluent using a small sprayer; the solution containing PBS only was taken as a control. Then, 10 cocoons were transferred into a mesh bag after drying, and they were buried in the soil. Until the next year, the cocoons (n = 30, per group) were removed from the soil before cutting them open, and the mortality of mature larvae was recorded. Larvae without any movement when they were touched gently using a fine brush were considered as dead. The SCP of *E. extimalis* was measured according to Lai et al. [13,15].

### 2.3. Effect of Different Temperature and Exposure Duration on Cold Resistance of E. extimalis

Three concentrations of the INA bacteria (*P. syringae* 1.3200) were used which were C1(0.1), C2(0.2), C3(0.3), and the OD at 600 nm was measured. For the bioassays, fresh oilseed rape siliques were dipped in each concentration of INA bacteria solutions for 1 min. The treated siliques were air-dried for 2 h at room temperature and then placed in a Petri dish. A total of 50 third-instar larvae were transferred to the siliques inside the dish, which was then placed in a growth chamber at 6 °C, 11 °C, 16 °C, and 21 °C with 70 ± 5% relative humidity. The siliques in the control sets were prepared using the same procedure as was previously described without the addition of any bacteria. All of the treatments were replicated four times. After 48 h of feeding on the treated siliques, the larvae were then fed daily with nontoxic siliques. On the 3rd, 7th, 11th, and 15th day, 30 larvae from each treatment group were removed to measure their SCP (n = 10), trehalose content (n = 5), and trehalose metabolic enzyme (n = 5), and gene expression (n = 10).

### 2.4. Measurement of Trehalase Activity

The trehalase activity was determined following modified protocols from Yang et al. [22] and Yu et al. [23]. The INA-treated and untreated insects were homogenized on ice for 10 min and 2 mL of PBS was added. The mixture was centrifuged at 4000× *g* for 15 min at 4 °C, and the supernatants were placed in fresh test tubes. Subsequently, 0.5 mL supernatant was transferred into a 15 mL test tube before adding 0.25 mL PBS (pH 5.8) and 0.5 mL trehalose standard solution (2 mmol/L). After their incubation at a constant temperature in humidity incubator for 60 min at 37 ± 1 °C, 2 mL 3,5-dinitrosalicylic acid reagent was added into the tube. Thereafter, the reaction was quenched by heating in a water bath for 5 min. The solution was then cooled under tap water for 3 min and diluted with deionized water to 10 mL. The absorbance was measured at 550 nm.

### 2.5. Quantification of TRE Expression after Cold Acclimation

A total of 10 *E. extimalis* larvae that were sampled from each treatment were used for total RNA isolation using a TRlzol Kit (Invitrogen, Carlsbad, CA, USA). The integrity of the RNA was examined using 1% agarose gels, and the concentration was analyzed using NanoDrop^TM^ 2000 spectrophotometer (Thermo Fisher Scientific, USA). To determine the Trehalase gene expression levels responding to the INA treatments at different concentrations and times, a quantitative real-time polymerase chain reaction (qRT-PCR) was performed using: forward primer 5’-CCACTGGACATCCCAAAGA-3’, reverse primer 5’-GGCTTCAGACCCGTCACAT-3’. Glyceraldehyde-3-phosphate dehydrogenase (GAPDH) (GenBank: MH790259) were chosen as the internal reference genes (forward primer 5’-CTGATGGCTGCCTTATTG-3’, reverse primer 5’-TTTGTCGGTGGTTGTGAA-3’). The cDNA synthesis SuperMix and TransScript ^®^ one—step gDNA removal (TransGen, Beijing, China) were used to synthesize the cDNA according to the manufacturer’s instructions. TransStart^®^ tip green qPCR SuperMix (TransGen) was used according to the manufacturer’s instructions to add the components of the qRT-PCR. This process was performed in LightCycler^®^480 (Roche, Basel, Swiss). The RT-qPCR was performed under the following conditions: 95 °C for 30 s, followed by 40 cycles of 95 °C for 5 s, and 60 °C for 31 s; melting curve: 95 °C for 15 s, 60 °C for 30 s, and 95 °C for 15 s. The obtained PCR products were detected using 1% agarose gel. RNase-free water was used as a negative control to replace the cDNA template. A serial 10-fold dilution of the cDNA temple was used to establish the standard curve for determining the amplification efficiency (E), which was calculated using the equation E = (10[−$\frac{1}{\mathrm{slope}}$]) × 100. Three technical and three biological replications were involved in this study, and the 2^−ΔΔCt^ method was for the statistical analysis. The fold-increase was obtained from the expression level of TRE genes in the INA bacteria-treated group divided by the water-treated control, and the ratio was used for expression profile analysis. One-way analysis of variance (ANOVA) and GraphPad Prism 6 (GraphPad Software, Inc., La Jolla, CA, USA) were used to analyze the expression levels of the Trehalase genes. In addition, a Tukey’s multiple comparisons test was used to evaluate the significant differences, and the results were displayed as mean ± SD.

## 3. Results 

### 3.1. Changes inSCP under INA Bacteria Stress

The bodies of the overwintering larvae that were poisoned by the INA bacteria became wizened and darker patches were present over the whole bodies when they were compared with the dead control larvae (Figure 1E,F). A general increase in the SCP after the INA bacteria application in the overwintering larva was evident (Figure 2). According to Figure 2, considerably lower values of the SCP were found in December, January, February, March, April, and May for the INA bacteria-treated larvae, even upon their exposure to the lower concentrations in comparison with those of the control, indicating that the application of the INA bacteria can significantly uplift the SCP of *E. extimalis* and decrease its cold resistance. However, there was no significant difference in the SCP between the samples when they were treated with the C1 and C3 concentrations. The SPCs of the larvae that were treated with *P. syringae* 1.3200, *P. syringae* 1.7277, and *E. pyrifoliae* 1.3333 at the C3 concentration were −10.72 °C, −13.73 °C, and −14.04 °C, respectively, in February 2015, which were two-folds greater than those that were observed in December 2014. The was no significant difference between the treatments with *P. syringae* 1.3200 and *P. syringae* 1.7277. The SCPs of the *P. syringae*-treated larvae were higher than those of the larvae that were treated with *E. pyrifoliae*. In addition, the SCPs of *E. extimalis* that were treated with the INA bacteria varied linearly with the temperature changes in the laboratory (data not shown). 

### 3.2. Changes in SCP under Different Concentrations and Temperatures

The treated larvae with *P. syringae* 1.3200 gradually lost their crawling capability and on each dying larvae, a darker patch appeared anteriorly. Finally, the bodies of the dead three-instar larvae became wizened and darker patches were present over the whole bodies when they were compared with the dead specimens under natural conditions (Figure 1B,C). The effects of the INA bacteria on the cold resistance of *E. extimalis* are depicted in Figure 3. After their exposure to the INA bacteria, the SCP of *E. extimalis* decreased initially and then increased. Additionally, the application of the INA bacteria significantly increased the SCP of *E. extimalis* when it was compared with non-treated samples on days 11 and 15. However, the microorganisms showed no obvious effects on the SCP at 16 °C and 21 °C except on days 3 and 7 (Figure 3). Among them, the SCPs of the treatments at 6 °C underwent the most obvious change. After being exposed to the INA bacteria for 3 days, the SCPs of *E. extimalis* were −48.60 °C, −44.3 °C, and −42.8 °C, which then increased to −2.61 °C, −5.12 °C, and −4.73 °C, respectively, over time. Similarly, the significant increases in the SCPs were detected at all of the concentrations in the larvae that were treated with the INA bacteria at 11 °C, except for the samples on day three.

### 3.3. Effect of Temperature and Exposure Duration on Trehalose Content 

The trehalose content was measured, and significant differences were found between the low and high temperatures. We have demonstrated that there were changes in the level of trehalose under the cold stress conditions in the treatment and control groups. The trehalose content was a function of the different durations of the application of *P. syringae* 1.3200, which was was evident (Figure 4). Overall, we found that the trehalose contents of *E. extimalis* were negatively correlated with the INA-bacteria concentrations and the detrimental effect that they had because of the microorganism application continued over time. In the environment that was warmed from 6–16 °C, a significant decrease in the trehalose content was observed at 16 °C (Figure 4). In the environment that was heated from 16–22 °C, an increase in the trehalose content was found at 22 °C, which was significantly lower than the content of it in the untreated group. The lowest level of trehalose content was observed after the exposure for 15 d at 16 °C, which was markedly lower than the content was in the control group. After the treatment, the trehalose content was almost as high as that in the control group from days 3 to 11, and this decreased to a low level at 15d, except for the content in the treatments at 22 °C. 

### 3.4. Effect of INA Bacteria on Trehalase Activity

The effects of the INA bacteria on the trehalase activity are depicted in Figure 5. Except for the INA-treated larvae at 22 °C, a substantial dose-dependent effect was generally observed when this was compared to the control group. Compared with untreated samples, the application of *P. syringae* 1.3200 obviously stimulated the trehalase activity. The trehalase activity increased with the increasing treatment concentration. After the treatment with *P. syringae* 1.3200 at concentrations of C1, C2, and C3 for 15 d at 11 °C, the trehalase activities of *E. extimalis* were 146.79 μg/g·min, 269.00 μg/g·min, and 284.88 μg/g·min, respectively. Considerably lower values of trehalase activity were found in the samples that were treated with the INA bacteria on days 3 and 7 at 6 °C and 11 °C, even upon their exposure to the lowest concentrations when they were compared with those of the control group. Higher values were observed on days 11 and 15 at 6 °C, 11 °C, and 16 °C. 

### 3.5. Expression Profiles of Trehalase Genes after Exposure to INA Bacteria

The upregulation of the TRE genes resulting from their exposure to the INA bacteria may be related to the cold resistance of *E. extimalis.* The larvae of *E. extimalis* were exposed to the C1, C2, and C3 concentrations of *P. syringae* 1.3200 at different temperatures and incubation times, and the expression levels of the trehalase genes in the larvae were monitored over 15 d (Figure 6). A marked dose-dependent effect was observed when this was compared to the control group. The results showed that the expression of the trehalase genes in *E. extimalis* gradually increased after their exposure to the C2 and C3 concentrations of the INA bacteria at 6 °C, 11 °C, 16 °C, and 22 °C within the incubation period. After they were treated with the INA bacteria at the C3 concentration on days 3, 7, 11 and 15, the TRE gene levels increased 15.51-fold, 62.88-fold, 105.13-fold, and 114.75-fold, respectively (Figure 6C). However, when they were treated with the lower concentrations of *P. syringae* 1.3200, the expression level increased first and then decreased, except for during the incubation at 16 °C (Figure 6A). The expression level of the TRE genes was found to increase with an increasing INA bacteria concentrations.

### 3.6. Phylogenetic Analysis

A phylogenetic tree was constructed which was based on the amino acid sequences from 14 species to examine the phylogenetic relationships between the *TRE* sequence of *E. extimalis* and its homologs in other insects. As shown in Figure 7, the *TRE* genes were most closely related to *Spodoptera frugiperda* (Lepidoptera: Noctuidae), indicating that these insects possessed a similar evolutionary development and physiological functions. 

## 4. Discussion

The supercooling point (SCP) is generally used to evaluate an insect’s cold tolerance because at this temperature, the insect’s bodies fluids spontaneously freeze [24,25]. The SCP represents the lowest lethal temperature for freeze-intolerant individuals. Therefore, the increase of the SCP can decrease the survival of the insect under low-temperature stresses. Previous studies have demonstrated that inoculating insects with INA bacteria can produce a remarkable reduction in their capacity to supercool [26,27,28]. Our studies have indicated that *E. extimalis* readily consumed various species of INA bacteria, which resulted in limiting the moths’ supercooling capacity to different temperatures and times. The supercooling capacity of the fed fennel borer increased by up to 4.4 °C in relation to the untreated samples. The effect of the INA bacteria application in the field setting might be even more remarkable given that the supercooling capacity in fully acclimatized, overwintering fennel borers may be substantially lower.

The cold tolerance of insects has long been a major issue in the Alpine region. The accumulation of cryoprotective solutes (polyols, sugars, and free amino acids) might contribute to the acquisition of cold tolerance in many insects, thereby improving the individual survival rates. Recent studies have reported that trehalose has high a concentration in the insect hemolymph, and it is the crucial ‘blood sugar’ that protects the cells against harsh environmental conditions [29,30,31]. Several studies have demonstrated the protective function of trehalose against potential harm that is induced by cold stress in overwintering insects. A marked increase in trehalose concentration was found in *Drosophila immigrans* after their recovery from cold stress conditions [16]. Wharton [32] has indicated that trehalose acts as a cryoprotectant in alpine weta (*Hemideina maori*) and alpine cockroach (*Celatoblatta quinquemaculata*) when they acclimate to rapid cold hardening. Our study showed that the application of INA bacteria affected the accumulation and utilization of trehalose in a stress-specific manner, and the findings indicate that trehalose accumulation can result in better survival in the treated samples. The water exclusion hypothesis can be used to explain the protective role of trehalose against cold stress. Trehalose molecules can preserve the membrane phospholipids and the native structure of proteins by replacing water at hydrogen bonding sites [17,33]. In comparison with bulk water, binding water with intracellular trehalose is less likely to freeze [34].

Trehalose, a non-reducing disaccharide that consists of two glucose molecules, is present in several organisms, including insects, plants, bacteria, fungi, and other invertebrates [29,35]. The only known pathway of cellular trehalose utilization is its hydrolysis by trehalase in insects [36]. To date, many genes encoding trehalose-6-phosphate synthase (TPS), soluble trehalase gene (Treh-1), and integral membrane trehalase gene (Treh-2) have been cloned from different insects and have shown to be involved in various physiological processes [22]. Our findings have demonstrated that the TRE genes were up-regulated following an INA bacteria application, resulting in an increased trehalase activity. Considering the negative correlation between trehalase and trehalose [37,38], the trehalose catabolism was stimulated. Consequently, the reduction in the trehalose content due to an INA bacteria application can result in poor survival rates in the winter season. More recently, trehalase has gained significance in pest management and prominence in contemporary studies because it plays a crucial role in carbon metabolism and enantiostasis [39]. Due to their high specificity to trehalose, most of the glycosidase inhibitors can competitively inhibit trehalase. Thus, some trehalose mimetics and analogs have been considered as potential insecticides, thereby placing trehalase on the list of biotechnologically important enzymes.

The introduction of pesticides in agriculture has not only increased crop productions, but it has also improved the quality of life for several people [40]. However, the indiscriminate use of dangerous chemical insecticides has led to pest resistance, human and animal poisoning, and environmental pollution [41,42]. Particularly in the Tibet Plateau, a relatively pristine region, the geographical conditions, thinner air, and lower temperatures may be regarded as reservoirs or sinks for atmospherically persistent chemical pesticides [43,44]. Many agricultural products contain pesticide residues, which have been linked to short- or long-term health effects in humans, including those that are teratogenic and carcinogenic [45]. Therefore, there is a pressing need in contemporary agricultural production for novel pesticides that do not adversely affect organisms other than the intended ones. In recent years, studies have demonstrated the possibility to control insect pests by utilizing INA bacteria, which show no direct noxious impacts on the normal development of insects aside from their SCPs and decreasing their hardiness to the cold, thereby causing the insects to freeze and die. Our results also showed that when they were compared with those of the control, the application of the INA bacteria considerably increased the SCP of *E. extimalis* (Figure 2). Therefore, the use of INA bacteria has the potential to become a new strategy for the biological control of pests. 

## Figures and Tables

**Figure 1 insects-13-00909-f001:**
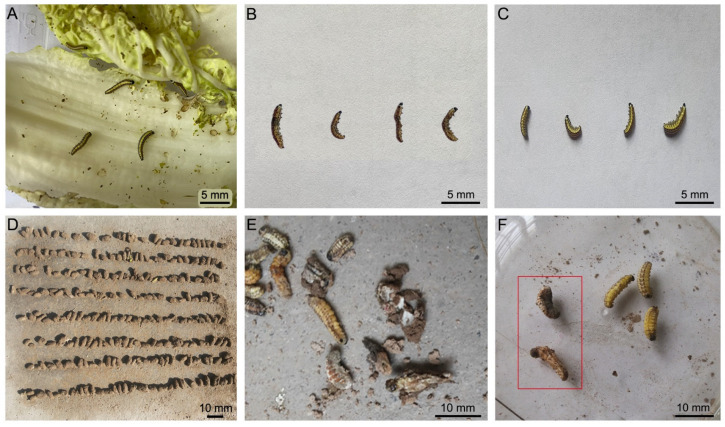
(**A**) Healthy larvae with uniform body color; (**B**) The bodies of dead third-instar larvae poisoned by INA bacteria; (**C**) The dead control larvae under natural condition; (**D**) Cocoons treated with INA bacteria before buried in the soil; (**E**) Overwintering larvae from the cocoons removed from the ground; (**F**) Dead overwintering larvae poisoned by INA bacteria as shown in a red box and dead control samples.

**Figure 2 insects-13-00909-f002:**
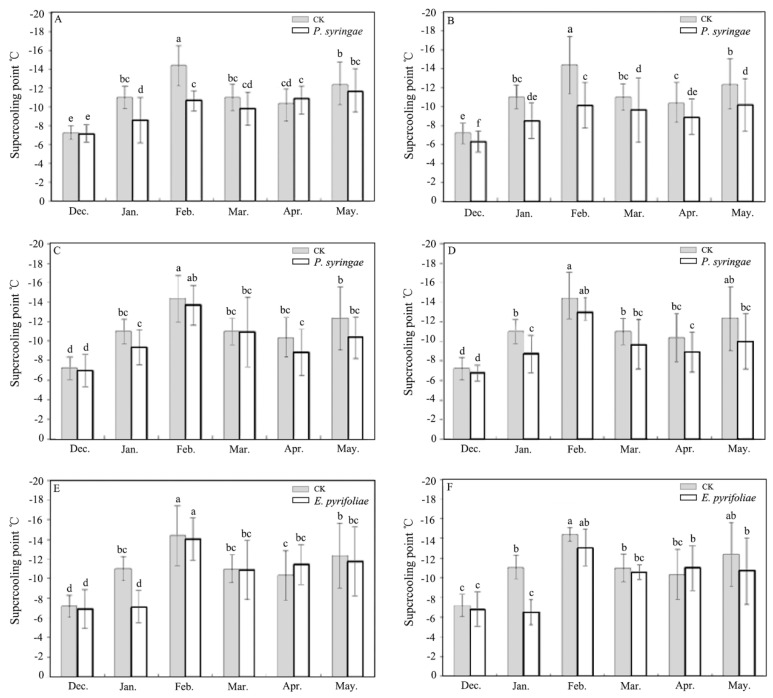
Effect of INA bacteria (*P. syringae* 1.3200; *P. syringae* 1.7277; *E. pyrifoliae* 1.3333) on the SCP of *E. extimalis* at different times. (**A**,**C**,**E**): bacterial concentration = 0.1 OD; (**B**,**D**,**F**): bacterial concentration = 0.3 OD. Results are shown as the mean ± SD. Letters on the error bars indicate significant differences analyzed using the one-way analysis of variance (ANOVA) with Tukey’s test (*p* < 0.05).

**Figure 3 insects-13-00909-f003:**
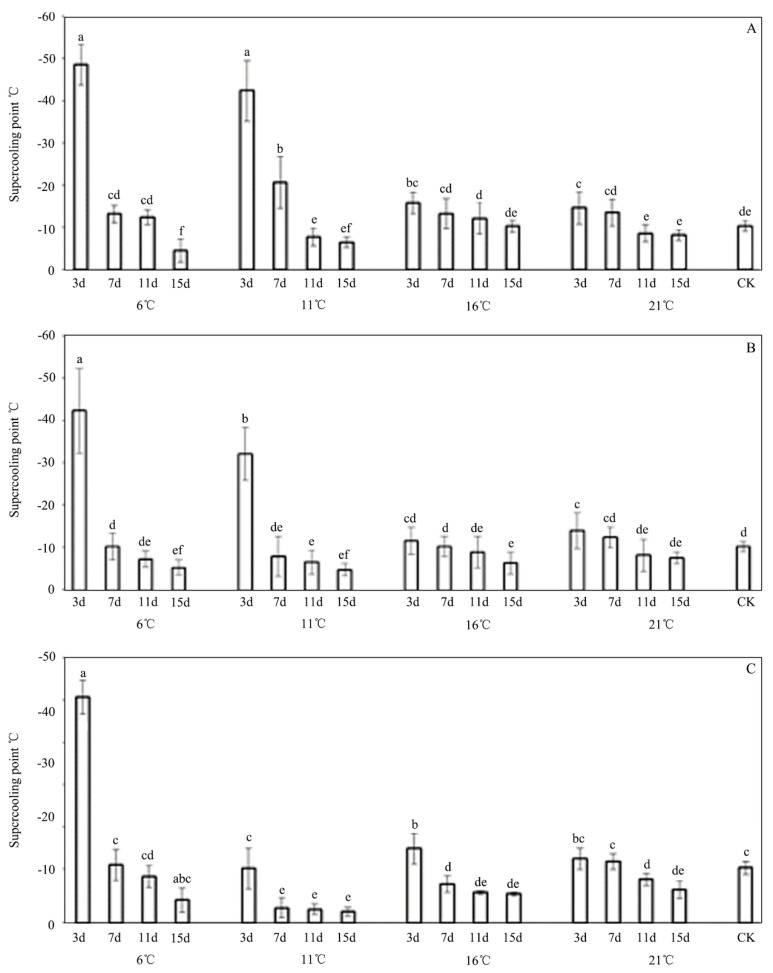
Supercooling point of *E. extimalis* treated with *Pseudomonas syringae*, 1.3200. (**A**) 0.1 OD; (**B**) 0.2 OD; (**C**) 0.3 OD. Results are shown as the mean ± SD. Letters on the error bars indicate significant differences analyzed by the one-way analysis of variance (ANOVA) with Tukey’s test (*p* < 0.05).

**Figure 4 insects-13-00909-f004:**
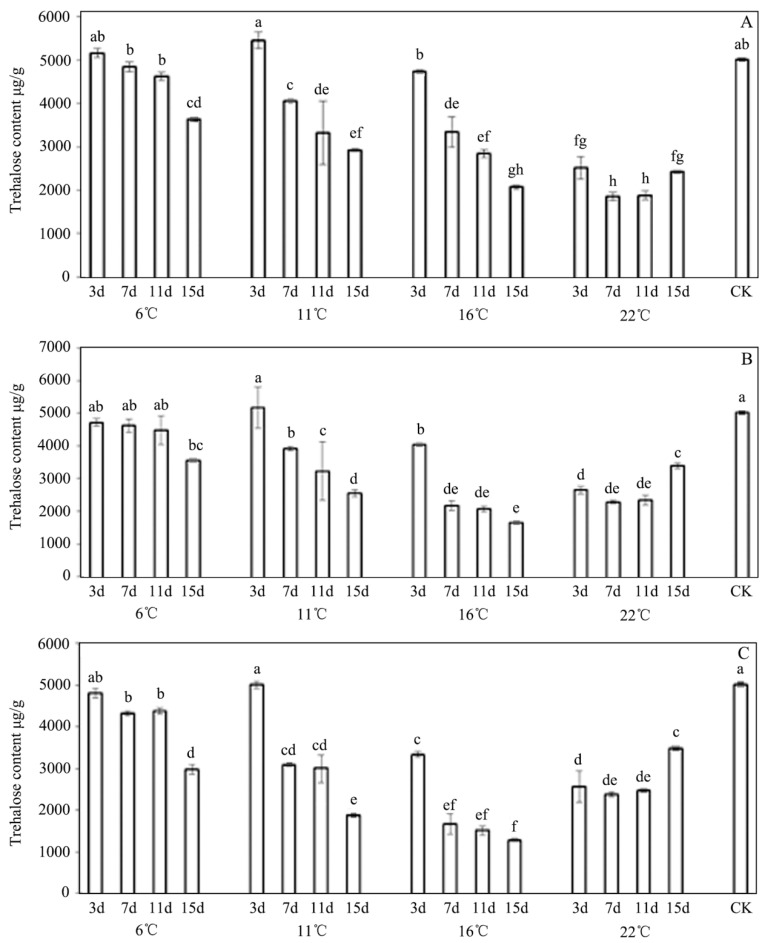
Effect of *P. syringae* 1.3200 concentration on the trehalose content of *E. extimalis*. (**A**) 0.1 OD; (**B**) 0.2 OD; (**C**) 0.3 OD. Results are shown as the mean ± SD. Letters on the error bars indicate significant differences analyzed using the one-way analysis of variance (ANOVA) with Tukey’s test (*p* < 0.05).

**Figure 5 insects-13-00909-f005:**
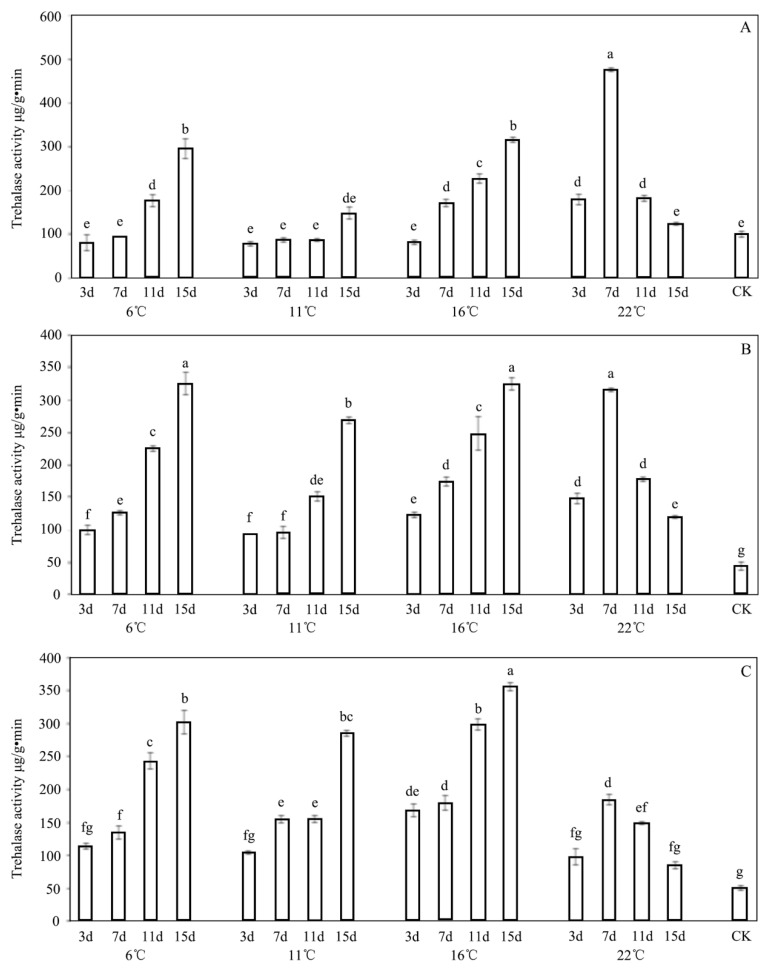
Effect of *P. syringae* 1.3200 concentration on trehalase activity of third-instar larvae of *E. extimalis*. (**A**) 0.1 OD; (**B**) 0.2 OD; (**C**) 0.3 OD. Results are shown as the mean ± SD. Letters on the error bars indicate significant differences analyzed by the one-way analysis of variance (ANOVA) with Tukey’s test (*p* < 0.05).

**Figure 6 insects-13-00909-f006:**
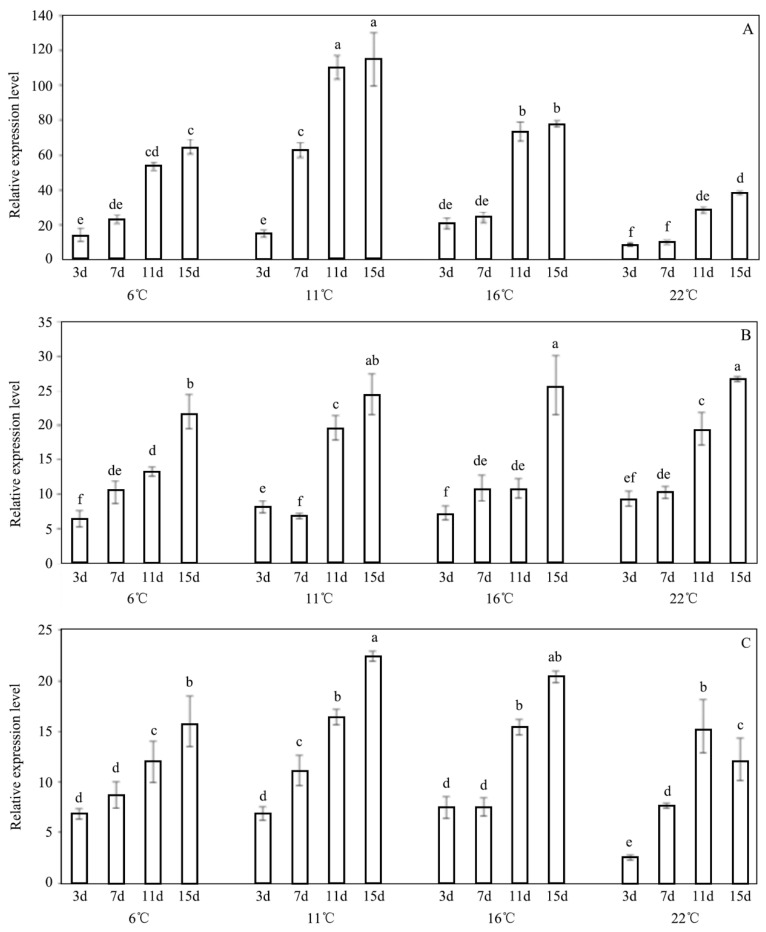
Expression profiles of trehalase genes of *E. extimalis* after exposure to INA bacteria. (**A**) 0.1 OD; (**B**) 0.2 OD; (**C**) 0.3 OD. Results are shown as the mean ± SD. Letters on the error bars indicate significant differences analyzed using the one-way analysis of variance (ANOVA) with Tukey’s test (*p* < 0.05).

**Figure 7 insects-13-00909-f007:**
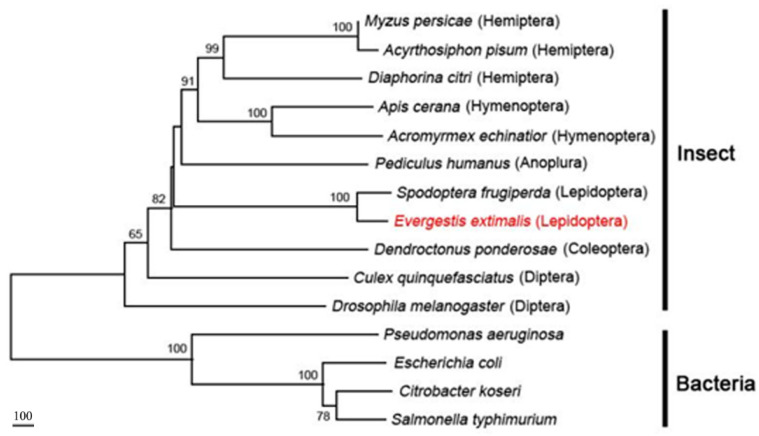
Phylogenetic analysis based on the amino acid sequence of trehalase. Bootstrap values for 1000 trials are indicated at each node.

## Data Availability

Data are contained within the article.

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
