# Peer review of "The Effect of Ice-Nucleation-Active Bacteria on Metabolic Regulation in Evergestis extimalis (Scopoli) (Lepidoptera: Pyralidae) Overwintering Larvae on the Qinghai-Tibet Plateau"

_insects, 2022, doi:10.3390/insects13100909_

Round 1
Reviewer 1 Report
Present manuscript “Effect of INA Bacteria on metabolic regulation in the overwintering larvae of Evergestis extimalis (Scopoli) (Lepidoptera: Pyralidae) on Qinghai-Tibet Plateau” by Hainan et al. Authors explained the effect of INA bacteria on Evergestis extimalis which is poorly written and explained. Before accepting this article, it needs major changes as follows:
1. English language is poor throughout the manuscript needs significant improvement preferably by a native speaker.
2. Many typos needs to be correceted like:
Abstract: Pseudomonas. syringae , there should not be period after genera name.
Methods: Cole to cold
Many more throughout the manuscript.
3. Method section is missing for INA bacteria effect in different months.
4. How you selected the doses of bacteria to be 0.1, 0.2 and 0.3 OD 600.
5. If the 3 ODs has been used throughout the experiments, provide the data of 0.3 DO600 for different months as well if not explain why not.
6. Not a single picture of the treated insects has been added into the manuscript.
I encourage authors to make a figure with the insect images from each section.
7. All the figure legends needs to be self-explanatory so improve them.
8. Discussion can be improved and more details of published data should be provided.
Author Response
Dear colleagues,
The manuscript " Effect of INA Bacteria on metabolic regulation in the overwintering larvae of Evergestis extimalis (Scopoli) (Lepidoptera: Pyralidae) on Qinghai-Tibet Plateau " has been carefully revised according to your comments and suggestion. Our point-by-point responses to your comments are list below.
Thanks again for your kind help!
With best wishes,
Hainan
Comments of Reviewer 1 and authors’ response:
Reviewer(s)' Comments to Author:
Reviewer: 1
Comments to the Author
Authors explained the effect of INA bacteria on Evergestis extimalis which is poorly written and explained. Before accepting this article, it needs major changes as follows:
Re: We appreciate the detailed and valuable comments and suggestions from you and the whole manuscript has been polished by a native speaker. Meanwhile, we have revised our manuscript according to your comments in the updated manuscript, which are highlighted using track-changes.
English language is poor throughout the manuscript needs significant improvement preferably by a native speaker.
Re: Thanks for your suggestion and comments. In order to make the writing more concise, clear and consistent, a native speaker helped us to polishing this manuscript.
Many typos needs to be correceted like:
Abstract: Pseudomonas. syringae , there should not be period after genera name.
Methods: Cole to cold
Many more throughout the manuscript.
Re: Thanks for your advice. We have revised our manuscript according to your comments in the updated manuscript, which are highlighted using track-changes.
Pseudomonas. syringae has been changed to Pseudomonas syringae.
Cole has been changed to cold.
Method section is missing for INA bacteria effect in different months.
Re: Thanks for your advice. The related text (effects of INA bacteria on Evergestis extimalis in exposure duration) has been added in the updated manuscript.
How you selected the doses of bacteria to be 0.1, 0.2 and 0.3 OD 600.
Re: Thanks for your question. The cells of Pseudomonas. syringae 1.7277, P. syringae 1.3200, and Erwinia pyrifoliae 1.3333 toward the end of the logarithmic phase were harvested by centrifugation at 15,000×g for 10min at 4℃. The cells were washed with PBS solution (pH 7.4) and were finally resuspended in the same solution. The number of cells in the suspension was counted using a Petroff Hausser counting chamber mounted on a phase-contrast microscope (BH-2, Olympus, Tokyo). The maximum concentration of the INA bacteria was approximate 1.5×108 CFU/mL (0.3 OD), and diluted the bacteria solutions to 1.0×108 CFU/mL (0.2 OD) and 5.0×107 CFU/mL (0.1 OD).
If the 3 ODs has been used throughout the experiments, provide the data of 0.3 DO600 for different months as well if not explain why not.
Re: Thanks for your question. We are ashamed of this mistake making you misunderstand. In our study, the bacterial resuspension was adjusted to 0.1 (C1), 0.2 (C2) and 0.3 (C3) OD at 600 nm, so 0.3 OD600 (C3) was used throughout the experiments rather than 3 OD.
Not a single picture of the treated insects has been added into the manuscript.
I encourage authors to make a figure with the insect images from each section.
Re: Thank you for your suggestion. Pictures (Figure 1) of the larvae treated with INA bacteria were added in the updated manuscript.
Discussion can be improved and more details of published data should be provided.
Re: Thank you for your question. A native Englishman has helped us to revise the discussion and the relevant references about insects were added to our manuscript.

Reviewer 2 Report
Hainan et al evaluated ice nucleation active 11 microbes (Pseudomonas. syringae 1.7277, P. syringae 1.3200, Erwinia pyrifoliae 1.3333) on E. 12 extimali. This work provides helpful information about new tools for the control of insect pests. The materials and methods section is adequate and in accordance with the reported results. However, before its publication, it would be convenient to carefully review all sections and improve grammar and syntax. In addition, the article would benefit from describing in more detail some of the result differences found in the study.
Abstract:
L10: Authors might use other connectors, for example, a short sentence that connects the problem with the alternative control (or a, therefore).
L11-12: This sentence is not clear, please consider rephrasing it.
Introduction: The introduction should be carefully reviewed;
L 27” Provide an approximate or exact amount, avoid the word tremendous which do not clearly indicate.
L 73: The word “whether” is missing in this sentence: …determine “whether” INA bacteria…
Materials and methods
L115: Rephrase this sentence, trehalose activity was determined following modified protocols from Zhong et al…
Results
L154: Consider describing the biological meaning of the statistical analysis, for example, significantly reduced, increased by two –fold…
L164: It is unclear; the authors should consider rephrasing the sentence, In addition, E. extimalis treated with INA bacteria followed the same trend..
L181: Rephrase the sentence.
L190: Rephrase the sentence.
L191: This sentence should be in the discussion.
L230: Delete “among them” and describe the type of change (e.g., increased?)
Discussion
L252: this sentence is not clear. It might be better “ previous studies …”However, the authors should beginning by discussing their results (no entendí que parte era tu sugerencia y que parte era la original de los autores).
L 256: This sentence is not clear, the authors should consider rephrasing it.
L 264: The comparison with Trehalose changes in mammals does not seem adequate in this context.
L277: there is no direct evidence for inferences about adaptation in this study.
Author Response
Dear colleagues,
The manuscript " Effect of INA Bacteria on metabolic regulation in the overwintering larvae of Evergestis extimalis (Scopoli) (Lepidoptera: Pyralidae) on Qinghai-Tibet Plateau " has been carefully revised according to your comments and suggestion. Our point-by-point responses to your comments are list below.
Thanks again for your kind help!
With best wishes,
Hainan
Comments of Reviewer 2 and authors’ response:
Reviewer: 2
Comments to the Author
This work provides helpful information about new tools for the control of insect pests. The materials and methods section is adequate and in accordance with the reported results. However, before its publication, it would be convenient to carefully review all sections and improve grammar and syntax. In addition, the article would benefit from describing in more detail some of the result differences found in the study.
Re: Thanks very much for your comments. The whole manuscript has been polished by a native speaker and we have revised our manuscript according to your comments, which are highlighted using track-changes. In addition, we have added some detailed results and comparisons between different groups in the updated manuscript.
Abstract:
L10: Authors might use other connectors, for example, a short sentence that connects the problem with the alternative control (or a, therefore).
Re: Thanks for your advice. “thus” has been changed to “therefore”.
L11-12: This sentence is not clear, please consider rephrasing it.
Re: Thanks for your question. We have revised the related text.
Introduction: The introduction should be carefully reviewed;
L 27” Provide an approximate or exact amount, avoid the word tremendous which do not clearly indicate.
Re: Thanks for your advice. We have revised the manuscript according to your comments. “3.5 million tonnes” has been used to replace “tremendous”.
L 73: The word “whether” is missing in this sentence: …determine “whether” INA bacteria…
Re: Thanks for your advice. We are ashamed of this mistake and “whether” has been added to the sentence.
Materials and methods
L115: Rephrase this sentence, trehalose activity was determined following modified protocols from Zhong et al…
Re: Thanks for your advice. This sentence has been revised.
Results
L154: Consider describing the biological meaning of the statistical analysis, for example, significantly reduced, increased by two –fold…
Re: Thanks for your advice. We have revised the related manuscript according to your comments.
L164: It is unclear; the authors should consider rephrasing the sentence, In addition, E. extimalis treated with INA bacteria followed the same trend.
Re: Thanks for your comments. We have revised the related text.
L181: Rephrase the sentence.
Re: Thanks for your comments. The related sentence has been revised.
L190: Rephrase the sentence.
Re: Thanks for your comments. We have revised the related text.
L191: This sentence should be in the discussion.
Re: Thanks for your comments. This sentence has been moved to the discussion.
L230: Delete “among them” and describe the type of change (e.g., increased?)
Re: Thanks for your comments. “among them” has been deleted and the related text has been revised.
Discussion
L252: this sentence is not clear. It might be better “ previous studies …”However, the authors should beginning by discussing their results (no entendí que parte era tu sugerencia y que parte era la original de los autores).
Re: Thank you for your comments and suggestions. We have revised the manuscript according to your comments.
L 256: This sentence is not clear, the authors should consider rephrasing it.
Re: Thanks for your advice. We have revised the related text.
L 264: The comparison with Trehalose changes in mammals does not seem adequate in this context.
Re: Thank you for your question. We are ashamed of this mistake. In our study, “blood sugar” represents the “hemolymph sugar” in insects and the related content has been revised.
L277: there is no direct evidence for inferences about adaptation in this study.
Re: Thank you for your question. We have deleted this sentence.
Round 2
Reviewer 1 Report
My all concerns has been addressed by the authors.